# Relationship between Atherogenic Dyslipidaemia and Lipid Triad and Scales That Assess Insulin Resistance

**DOI:** 10.3390/nu15092105

**Published:** 2023-04-27

**Authors:** Hernán Paublini, Angel Arturo López González, Carla Busquets-Cortés, Pilar Tomas-Gil, Pere Riutord-Sbert, José Ignacio Ramírez-Manent

**Affiliations:** 1Research Group Adema Salud IUNICS, 07003 Palma, Spain; 2Faculty of Dentistry, University School ADEMA, 07009 Palma, Spain; 3IDISBA, Balearic Islands Health Research Institute Foundation, 07004 Palma, Spain; 4General Practitioner Department, Balearic Islands Health Service, 07003 Palma, Spain; 5Faculty of Medicine, University of the Balearic Islands, 07009 Palma, Spain

**Keywords:** atherogenic dyslipidaemia, lipid triad, insulin resistance, atherogenic risk

## Abstract

Background: Atherogenic dyslipidaemia (AD) and lipid triad (LT) are characterised by high triglyceride levels together with low HDL and normal or high LDL cholesterol and are favoured by a persistent state of insulin resistance (IR), which increases the release of free fatty acids from abdominal adipose tissue. This alteration in the lipid profile favours the accelerated development of atherosclerosis, which is the most important cause of morbidity and mortality in all countries in the developed and developing world. One of the elements that plays a major role in the genesis of AD is IR. The aim of this study was to determine the relationship between variables that assess atherogenic risk (AD and LT) and scales that assess the risk of presenting insulin resistance. Methods: A descriptive cross-sectional study of 418,343 workers was conducted to evaluate atherogenic dyslipidaemia and lipid triad; a relationship with three insulin resistance risk scales (Triglycerides/HDL, TyG index, METS-IR) was established. The usefulness of IR risk scales for predicting AD and LT was calculated by applying ROC curves, obtaining the area under the curve (AUC) and cut-off points with their sensitivity, specificity, and Youden index. Multivariate analysis was performed by binary logistic regression. Results: The prevalence of high-risk values for insulin resistance with all of the scales is much higher in people with AD and LT compared to those without. The ROC curves present us with an AUC with the three insulin resistance risk scales for the two dyslipidaemias studied with figures ranging between 0.856 and 0.991, which implies that the results are good/very good. Conclusions: A relationship between atherogenic dyslipidaemia and the three insulin resistance risk scales assessed is revealed, with higher IR mean values and prevalence in people with atherogenic dyslipidaemia and lipid triad. The three scales make it possible to adequately classify the presence of AD and LT. The highest AUC is presented by the triglycerides/HDL scale, with a result close to 1. METS-IR is the most recommended formula to estimate insulin resistance.

## 1. Introduction

Atherogenic dyslipidaemia (AD) [1] and lipid triad (LT) [2,3] are characterised by high triglyceride levels together with low HDL and normal or high LDL cholesterol [4,5]. This lipid alteration is common in metabolic syndrome and type 2 diabetes mellitus [6] and is favoured by a persistent state of insulin resistance (IR) [7,8], which increases the release of free fatty acids from abdominal adipose tissue. These fatty acids accumulate in the liver and form part of triglyceride-rich VLDL cholesterol. When VLDL leaves the liver, it exchanges triglycerides for esterified cholesterol with the other lipoproteins (HDL and LDL), thus becoming triglyceride-rich lipoproteins, which increases their atherogenic power [9]; alteration in the lipid profile favours the accelerated development of atherosclerosis [10], which is the most important cause of morbidity and mortality in all countries in the developed and developing world [11,12].

This increase in atherogenesis is closely related to current lifestyles [13] in which there is a high consumption of processed and high-calorie foods (soft drinks, fried foods, and large amounts of carbohydrates) together with a large percentage of sedentary lifestyles [14], which has led to an alarming increase in the prevalence of obesity and cardiometabolic disorders [15]. These include metabolic syndrome, obesity, non-alcoholic fatty liver disease (NAFLD), and the development of early cardiovascular disease [6]—as fatty liver and liver damage result in a fundamental disturbance of fatty acid metabolism—with increased lipid triad and atherogenic dyslipidaemia, affecting cardiovascular health [16].

As mentioned above, one of the elements that plays a major role in the genesis of AD is IR [3,17,18,19,20,21,22]. There are several types of lipoproteins: high-density lipoproteins (HDL) and low-density lipoproteins (LDL) that transport mainly cholesterol [23], and very low-density lipoproteins (VLDL) [24] that transport triglycerides. In normal situations, the metabolism of triglycerides in the liver is regulated by insulin, which activates lipoprotein lipase (LPL), but in states of IR [25]—such as type 2 diabetes mellitus, metabolic syndrome, and obesity—insulin deficiency decreases LPL activity, which implies that there is a greater amount of free fatty acids in the blood that will allow greater synthesis of triglycerides in the liver, which will result in an excess of VLDL, which are particles that remain in the blood longer and thus increase their atherogenic power [26,27].

The aim of this study was to determine the relationship between variables that assess atherogenic risk (atherogenic dyslipidaemia and lipid triad) and scales that assess the risk of presenting insulin resistance.

## 2. Methods

A descriptive, cross-sectional study was conducted on 418,343 workers (172,282 women and 246,061 men) from different regions in Spain belonging to different labour sectors, essentially hospitality, construction, commerce, health, public administration, transport, education, industry, and cleaning. The study was carried out between January 2019 and June 2020. The people who participated in the study were chosen from those who attended the health examinations carried out in the occupational health medical services of the different companies.

### 2.1. Inclusion Criteria

-Being over 17 years of age and under 70.-Working in one of the companies served by the occupational health services participating in the study.-Accepting the study conditions and their participation in it.-The PRISMA flow chart is shown in Figure 1.

### 2.2. Determination of Variables

The different variables, both anthropometric and clinical or analytical, were determined by the health professionals of the different companies. To avoid interobserver bias, measurement processes were standardised.

Waist circumference was measured with the person in a standing position, with their feet together and abdomen relaxed, placing the tape measure parallel to the floor at the middle height between the last rib and the hip.

Blood pressure was measured while the person was seated and after a rest of at least 10 min. Three determinations were made and the mean of the three was obtained. An OMRON M3 sphygmomanometer was used for this determination.

Analytical parameters were obtained by venepuncture after prolonged fasting (at least 12 h) by applying different techniques: enzymatic for cholesterol, triglycerides, and glycaemia, and precipitation with dextran-sulphate Cl2Mg for HDL. LDL values were obtained by indirect methods applying the Friedewald formula. All parameters were expressed in mg/dL.

Different insulin resistance risk scales were calculated:-Triglycerides/HDL-c. Ratio above 2.4 is considered high risk [28].-Triglyceride-glucose index (TG index) [29] obtained by applying the following formula: Ln(Triglycerides [mg/dL] × glucose [mg/dL]/2). Values above 8.8 are considered high.

Other parameters were obtained from the TyG index: TyG-BMI [30], TyG-waist [30], and TyG-WtHR [31] (waist to height ratio).

Metabolic score of insulin resistance (METS-IR) [32] METS-IR = Ln((2 × Glucose) + Triglycerides) × BMI)/(Ln(HDL-c)). High values are considered as 50 and over.

Atherogenic dyslipidaemia was considered when triglycerides presented values from 150 mg/dL and HDL was less than 40 mg/dL in men and less than 50 mg/dL in women, with LDL values less than 160 mg/dL. When LDL values above 160 mg/dL were found, it was considered a lipid triad [33].

A smoker was considered to be a person who had consumed a daily cigarette (or any tobacco equivalent) in the previous month, or who had quit smoking less than 12 months before.

Social class was obtained from the 2011 National Classification of Occupations (CNO-11) and applying the criteria of the Spanish Society of Epidemiology [34]. It was stratified into three categories: Class I. Includes managers, university professionals, athletes, and artists. Class II. Skilled workers in intermediate occupations and self-employed workers. Class III. Includes unskilled workers.

### 2.3. Ethical Considerations and Aspects

All the processes of this study were governed by the ethical standards of the institutional research committee and the 2013 Declaration of Helsinki. Anonymity and confidentiality of the data collected were guaranteed at all times. The Balearic Islands Research Ethics Committee (CEI-IB) approved the study, which was obtained with the following indicator: IB 4383/20. The data of each participant were coded, and only the person responsible for the study could know the identity of each person. The research team is committed to strict compliance with Organic Law 3/2018, of December 5, on the protection of personal data and guarantee of digital rights, guaranteeing the participants in this study the exercise of the rights of access, rectification, cancellation, and opposition of the data collected.

### 2.4. Statistical Analysis

For quantitative variables, the mean and standard deviation were calculated, and Student’s *t*-test was applied. The chi-square test was used for qualitative variables. The usefulness of the insulin resistance risk scales for predicting atherogenic dyslipidaemia and lipid triad was calculated by applying ROC curves, obtaining the area under the curve (AUC) and the cut-off points with their sensitivity, specificity, and Youden index. Multivariate analysis was performed by binary logistic regression. Statistical analysis was performed with the SPSS 28.0 program, with *p* < 0.05 as the accepted level of statistical significance.

## 3. Results

Table 1 shows that almost 59% of the sample were men. The average age is slightly over 40 years, with the majority between 30 and 49 years of age (58.8%). Three out of four people belonged to social class III and one in three smoked. The clinical and analytical parameters show more unfavourable values in men.

Table 2 shows that the mean values of the insulin resistance risk scales analysed are higher in persons with atherogenic dyslipidaemia and lipid triad in both sexes.

Table 3 shows that the prevalence of high-risk values for insulin resistance with any of the scales is much higher in people with atherogenic dyslipidaemia and lipid triad compared to those without.

Figure 2 and Table 4 show the results of the ROC curves. The highest areas under the curve were found for the triglyceride/HDL scale with values very close to 1.

In Table 5 we can see the relationship between fasting blood glucose and different lipid levels. For this, we have divided the glycemia figures by sex and stratified them into three groups, which correspond to normal figures below 100 mg/dL, which could correspond to prediabetes; figures between 100 and 125 mg/dL; and figures for above 126 mg/dL, which would correspond to diabetes mellitus [35].

In Table 6 we assess the relationship between waist circumference and the three evaluated formulas for the risk of insulin resistance. In our multivariate analysis using binary logistic regression, we found results with a high significant value in the three formulas studied with very narrow confidence intervals. With the highest OR for the METS-IR formula.

## 4. Discussion

Our study population is made up of 418,343 people, 41% of whom are women (172,282) and 59% men (246,061), aged between 18 and 69 years. Almost 60% of the sample is in the age range between 30 and 49 years, with most of them belonging to social class III.

Belonging to a social class with low purchasing power is associated with difficulty in acquiring healthier foods and therefore a larger intake of saturated fats, sugars, and foods with a high caloric content but low nutritional value [36,37,38]. This situation is associated with a higher rate of overweight and obesity, which in turn is linked to a greater prevalence of dyslipidaemia and insulin resistance [39,40,41]. This could be assessed as a bias in our study. However, since our objective was to determine the relationship between atherogenic dyslipidaemia and the lipid triad with the scales that assess the risk of presenting insulin resistance, we consider that it is an ideal population to carry out this study.

From the different accepted insulin resistance scales, we have selected Triglycerides/HDL-c, TG index, and METS-IR. There are other recognized methods to determine insulin resistance, such as QUICKI, HOMA-IR, and Matsuda index; however, it is necessary to know the fasting insulin level [42], which we did not have in our study and is also very difficult to obtain in primary care doctors’ offices. Of the non-insulin-based formulas, we have selected those that have shown a better correlation with insulin resistance [43,44].

In the three scales of risk of insulin resistance that we analysed, we found that its average values were higher in both people with atherogenic dyslipidaemia and lipid triad. In addition, these results have a statistical significance in both men and women. In our bibliographic search, we found several articles that relate atherogenic dyslipidaemia to insulin resistance. One studied proprotein converts subtilisin/kexina type 9 (PCSK9), finding a relationship between this protein and atherogenic dyslipidaemia and insulin resistance [45]. Another studied the relationship between insulin resistance and metabolic syndrome, relating it to early atherosclerosis. Describing a development of atherogenic dyslipidaemia in patients with insulin resistance, this article mentions the possibility that insulin resistance facilitates the formation of small and dense particles of low-density lipoproteins as a part of its metabolic transition. These particles are much more atherogenic and are part of the so-called atherogenic lipid triad [46]. At present, it has become a topic of great concern that different authors have worked on [47,48,49,50].

When we assess the prevalence of high values of insulin resistance in relation to the presence or absence of atherogenic dyslipidaemia and lipid triad, we find that the prevalence of high-risk values for insulin resistance is much greater in people who present these lipid alterations [51,52,53].

When we assessed the three insulin resistance risk formulas separately, we found that in one of them (Triglycerides/HDL-c high), the prevalence of insulin resistance risk values is 100% for both atherogenic and lipid triad dyslipidaemia, and that this follows the same pattern in both sexes. There is a positive relationship between the TG/HDL-c ratio and the incidence of type 2 diabetes, regardless of initial insulin resistance [54,55]. In addition, the TG/HDL-c ratio has been shown to be useful in predicting total mortality, fatal and non-fatal cardiovascular events, regardless of age, race, smoking, hypertension, diabetes, and severity of coronary artery disease [56,57,58].

Multiple studies have been published trying to establish the cut-off point for the TG/HDL-c index. Some authors establish the cut-off point of high risk at >3 [59], other works mark the high risk from >4 [60], and in other publications, the index is established >2.4 [61,62], or even lower [63], without having been able to establish up to now what the index should be [64]. Being aware that the atherogenic index is related to the number of small LDL particles and that a TG/HDL-c index >2 indicates a greater number of small LDL particles [65], in our work, we have used >2.4 as the high risk cut-off point, which is the cut-off point of other studies carried out in the Hispanic population [59,60] and the cut-off point for the Spanish population recommended by our scientific societies, semFYC (Spanish Society of Medicine Family and Community), SEMERGEN (Spanish Society of Primary Care Physicians), SEMG (Spanish Society of General Medicine), and SEA (Spanish Society of Arteriosclerosis) [66]. Below 2.4, we could consider moderate risk.

The second of the IR formulas (TyG index high) also shows very high insulin resistance values for this population, with close to 100% in both men and women and for both types of dyslipidaemia.

The third formula that we assessed (METS-IR high) also presents a much higher prevalence in the population with the presence of atherogenic dyslipidaemia and lipid triad than in the population without these dyslipidaemias. However, the prevalence percentage is lower than with the other two formulas, and in this case, we do observe differences between men and women, although the prevalence of insulin resistance risk in women is much higher in the groups with dyslipidaemia than in those without it: 38.6% for atherogenic dyslipidaemia and 35.1% for lipid triad. In men, these percentages are two-fold, reaching 62.2% in the former and 62.3% for the lipid triad. These differences between men and women have been described in other studies and are considered to be due to sex hormones and adipokines that cause greater insulin sensitivity in women [67,68,69,70].

This could be due to the different components that are part of the formulas for calculating the risk of insulin resistance. In the first one, only lipids are considered. In the second formula, blood glucose is added as one of the components to be assessed. In the last of the evaluated formulas, in addition to lipids and glycaemia, BMI is added. It is in this formula where the percentages are lowest. This seems logical, since what we are looking for is the association of IR with dyslipidaemia parameters, and when introducing other elements in the formulas, these can be diluted. Regarding these formulas, different studies find that the highest predictive value of insulin resistance is offered by the METS-IR, although it has not been evaluated for the atherogenic dyslipidaemia and lipid triad [71,72,73,74].

In the analysis of the ROC curves, the AUC obtained with the three insulin resistance risk scales ranged between 0.905 and 0.964 for atherogenic dyslipidaemia in men, which is interpreted as a good to very good result, and with a very similar result in women. In these, it should be noted that when calculating the risk of insulin resistance using the TG/HDL formula, the AUC is almost 1, which is an excellent result and an almost perfect diagnostic value.

When evaluating the AUC of the different insulin resistance risk scales with the lipid triad, the results we obtain are very similar to those obtained for atherogenic dyslipidaemia, in both men and women, which implies that the result is also good, very good.

A cut-off of 4 was established for men in the TG/HDL formula, with a Youden index of 0.871 in atherogenic dyslipidaemia and 0.780 in the lipid triad. This indicates a high specificity and sensitivity. In the case of women, the cut-off point was established at 3 for both dyslipidaemias, with a Youden index close to 1. This indicates that the test is almost perfect.

With the second of the formulas, TyG index, the cut-off point obtained for men is 8.9 in both atherogenic dyslipidaemia and lipid triad, with a Youden index greater than 0.7 in both parameters, which implies a great sensitivity and specificity in its relationship with this scale for evaluating insulin resistance. In the case of women, the cut-off point was established at 8.7 for atherogenic dyslipidaemia and 8.8 in the lipid triad, with a very high Youden index in both cases—0.879 in atherogenic dyslipidaemia and 0.840 in the lipid triad. This implies, in the latter case, a probability of 84% that there really is a relationship between the lipid alteration and the insulin resistance scale.

In the last of the formulas used, METS-IR, we obtained a cut-off point of 44.5 and 45 in men, and 40 in women for both lipid alterations, although with this formula, the Youden index is lower than in the previous ones—less than 0.6 in women and slightly higher than 0.6 in men. When evaluating these results together with the AUC of the different variables, the lowest value being 0.856 in LT for women and the highest value being 0.905 in AD for men, we can conclude that this scale also has a high predictive value for both dyslipidaemias.

In the same way that we have evaluated the inclusion of lipids in insulin resistance formulas, we have considered it interesting to evaluate the relationship between the different lipid fractions of plasma and fasting glycemia, since the transformation of excess glucose into lipids has been demonstrated, as well as the relationship of elevated glucose levels with diabetes mellitus and insulin resistance [75].

In our work, we found that as total cholesterol figures increase from below 200 mg/dL to more than 240 mg/dL, LDL-c from below 130 mg/dL to more than 160 mg/dL, or Triglycerides from less than 150 mg/dL to more than 200 mg/dL, fasting blood glucose levels increase in both men and women, which agrees with other published studies [76,77]. This coincides with the results obtained, since both triglycerides and fasting glucose intervene in the TyG index and the METS-IR formula. The latter is already recognized as a useful tool to assess metabolic health in primary care [32].

The influence of lipids on the development of arteriosclerotic disease is known [78], which produces an inflammation of the arterial wall with accumulation of lipids in it [79].

The presence of elevated plasma levels of LDL-c constitutes a cardiovascular risk factor [80]. Diabetes mellitus has also been shown to accelerate the development of cardiovascular disease through hyperglycemia, which, together with elevated LDL-c levels, leads to increased oxidative stress and inflammation [81].

In the patient with diabetes, the change in the serum lipid profile causes a greater production of triglycerides in the liver. At the same time, the lack of insulin causes an increase in the production of ROS (reactive oxygen species) and chronic inflammation that consequently accelerates the development of arteriosclerosis and cardiovascular disease [82]. These effects are favoured by insulin resistance [83].

In our work, we can see how there is an association between the increase in blood lipids and fasting serum glucose levels. Similarly, we found an association between atherogenic dyslipidaemia and lipid triad with the resistance risk scales evaluated.

The mean values and prevalence of elevated values of all insulin resistance risk scales are higher in persons with atherogenic dyslipidaemia and lipid triad. The high values of the areas below the curve in the ROC curves enable us to affirm that the three IR risk scales analysed have a high predictive value for AD and LT.

Given the results obtained, we decided to carry out a multivariate analysis using binary logistic regression, establishing the IR determined by waist circumference as the dependent variant. We published a previous study in which this parameter was the one that best predicted IR [84]. Additionally, as covariates, the three IR scales that we have evaluated in this study (TG/HDL, TyG index, and METS-IR), controlled for age and sex. The results obtained show that although all the independent variables present a good estimate of the appearance of IR (OR greater than 1), the one that shows a higher result is METS-IR, which would imply that it is the best formula to estimate the IR.

The novelty of this study is that it is the first to establish a relationship between insulin resistance risk scales and atherogenic dyslipidaemia and lipid triad, in addition to being the first to establish cut-off points for predicting atherogenic dyslipidaemia and lipid triad.

## 5. Strengths and Limitations

The main strengths of our research are the large sample size, which exceeds 418,000 workers, and the number of insulin resistance risk scales analysed.

The main limitation is that insulin resistance was not determined by objective tests but by risk scales.

## 6. Conclusions

There are higher mean values and prevalence of elevated values for all insulin resistance risk scales in persons with atherogenic dyslipidaemia and lipid triad.

In the ROC curves, we observed that the three insulin resistance risk scales enable us to adequately classify the presence of atherogenic dyslipidaemia and lipid triad, with the highest areas under the curve for the triglyceride/HDL scale.

Of the three scales evaluated, the one with the best results is the METS-IR, and therefore it would be the most recommended formula for estimating insulin resistance.

## Figures and Tables

**Figure 1 nutrients-15-02105-f001:**
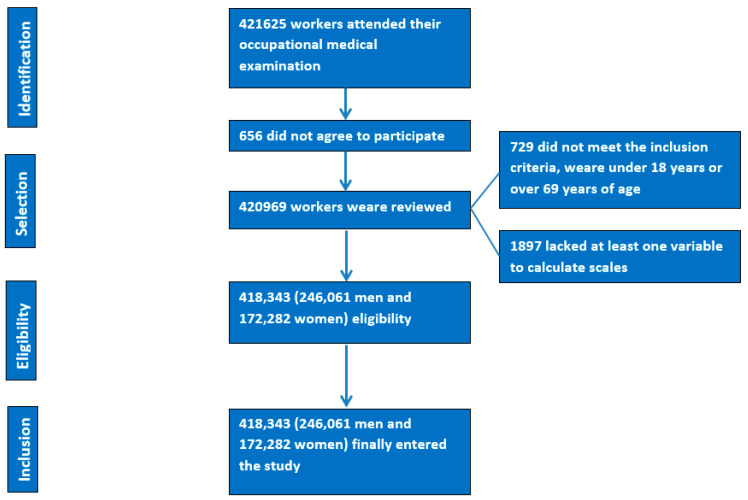
PRISMA flow chart of the participants in the study.

**Figure 2 nutrients-15-02105-f002:**
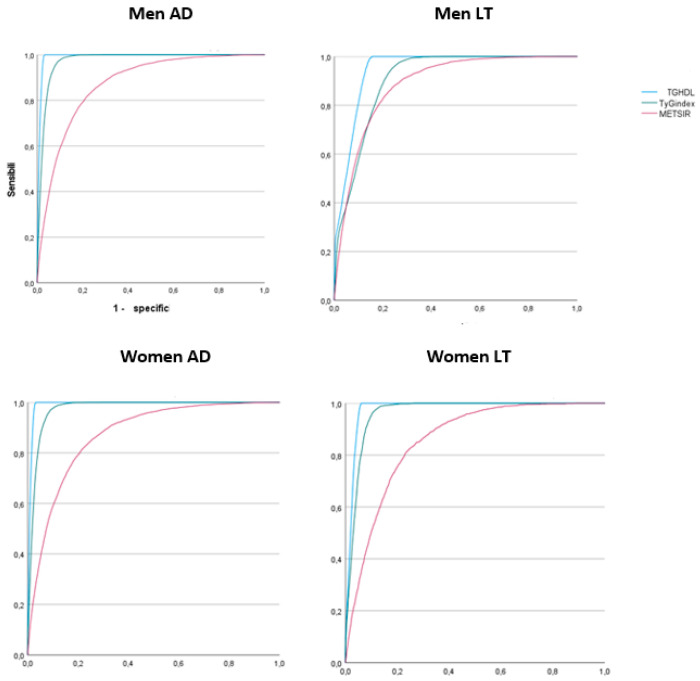
ROC curve. AD atherogenic dyslipidaemia. LT lipid triad.

**Table 1 nutrients-15-02105-t001:** Characteristics of the population.

	Women *n* = 172,282	Men *n* = 246,061	Total *n* = 418,343	
	Mean (SD)	Mean (SD)	Mean (SD)	*p*-Value
Age (years)	39.6 (10.8)	40.6 (11.1)	40.2 (11.0)	<0.0001
Height (cm)	161.8 (6.5)	174.6 (7.0)	169.4 (9.3)	<0.0001
Weight (kg)	66.2 (14.0)	81.4 (14.7)	75.1 (16.2)	<0.0001
Waist circumference (cm)	74.8 (10.6)	86.2 (11.1)	81.5 (12.2)	<0.0001
SBP (mmHg)	117.4 (15.7)	128.2 (15.5)	123.7 (16.5)	<0.0001
DBP (mmHg)	72.6 (10.4)	77.8 (11.0)	75.6 (11.0)	<0.0001
Total cholesterol (mg/dL)	190.6 (35.8)	192.6 (38.9)	191.8 (37.7)	<0.0001
HDL-c (mg/dL)	56.8 (8.7)	50.3 (8.5)	53.0 (9.1)	<0.0001
LDL-c (mg/dL)	116.1 (34.8)	118.0 (36.7)	117.2 (35.9)	<0.0001
Triglycerides (mg/dL)	89.1 (46.2)	123.7 (86.4)	109.5 (74.6)	<0.0001
Glycaemia	87.8 (15.1)	93.3 (21.3)	91.0 (19.2)	<0.0001
ALT (U/L)	20.2 (13.6)	31.0 (20.2)	26.6 (18.6)	<0.0001
AST (U/L)	18.2 (7.9)	24.4 (13.3)	21.7 (11.7)	<0.0001
GGT (U/L)	20.4 (19.7)	35.8 (39.3)	29.6 (33.6)	<0.0001
	**%**	**%**	**%**	** *p* ** **-value**
18–29 years	20.7	18.8	19.6	<0.0001
30–39 years	29.7	27.6	28.4	
40–49 years	29.6	30.1	29.9	
50–70 years	20.0	23.6	22.2	
Social class I	6.9	4.9	5.7	<0.0001
Social class II	23.4	14.9	18.4	
Social class III	69.7	80.3	75.9	
Non-smokers	67.2	66.6	66.9	<0.0001
Smokers	32.8	33.4	33.2	

SBP systolic blood pressure. DBP diastolic blood pressure. HDL-c high-density lipoprotein-cholesterol. LDL-c low-density lipoprotein-cholesterol. AST aspartate transaminase. ALT alanine transaminase. GGT gamma-glutamyl transferase. A value of *p* < 0.05 was accepted as significant.

**Table 2 nutrients-15-02105-t002:** Mean values of insulin resistance risk scales according to the presence or absence of atherogenic dyslipidaemia and lipid triad by sex.

	**Women**			**Men**		
	**Non AD *n* = 165,431**	**Yes AD *n* = 6851**		**Non AD *n* = 227,030**	**Yes AD *n* = 19,031**	
	**Mean (SD)**	**Mean (SD)**	** *p* ** **-Value**	**Mean (SD)**	**Mean (SD)**	** *p* ** **-Value**
Triglycerides/HDL	1.5 (0.7)	4.5 (1.9)	<0.0001	2.3 (1.6)	6.5 (3.4)	<0.0001
TyG index	8.1 (0.4)	9.2 (0.4)	<0.0001	8.4 (0.5)	9.3 (0.4)	<0.0001
METS-IR	34.6 (7.9)	48.2 (9.7)	<0.0001	38.1 (7.5)	52.7 (8.7)	<0.0001
	**Non LT *n* = 170,566**	**Yes LT *n* = 1716**	** *p* ** **-value**	**Non LT *n* = 240,669**	**Yes LT *n* = 5392**	** *p* ** **-value**
Triglycerides/HDL	1.6 (0.9)	5.1 (3.1)	<0.0001	2.5 (1.8)	8.0 (5.3)	<0.0001
TyG index	8.1 (0.5)	9.2 (0.5)	<0.0001	8.5 (0.6)	9.5 (0.6)	<0.0001
METS-IR	35.0 (8.3)	47.6 (9.3)	<0.0001	39.0 (8.3)	53.2 (9.2)	<0.0001

TG/HDL triglycerides/high-density lipoprotein. TyG triglyceride glucose index. METS-IR Metabolic score for insulin resistance. AD atherogenic dyslipidaemia. LT lipid triad. A value of *p* < 0.05 was accepted as a significant difference.

**Table 3 nutrients-15-02105-t003:** Prevalence of high values of insulin resistance risk scales according to the presence or absence of atherogenic dyslipidaemia and lipid triad by sex.

	**Women**			**Men**		
	**Non AD *n* = 165,431**	**Yes AD *n* = 6851**		**Non AD *n* = 227,030**	**Yes AD *n* = 19,031**	
	**% (95% CI)**	**% (95% CI)**	** *p* ** **-Value**	**% (95% CI)**	**% (95% CI)**	** *p* ** **-Value**
Triglycerides/HDL high	14.4 (14.4-4.4)	100.0 (100.0-100.0)	<0.0001	18.8 (18.8-18.9)	100.0 (100.0-100.0)	<0.0001
TyG index high	9.0 (9.0-9.0)	96.9 (96.2-97.6)	<0.0001	21.6 (21.6-21.6)	95.7 (95.1-96.3)	<0.0001
METS-IR high	5.1 (5.1-5.1)	38.6 (37.9-39.3)	<0.0001	7.1 (7.1-7.1)	62.2 (61.8-62.6)	<0.0001
	**Non LT *n* = 170,566**	**Yes LT *n* = 1716**	** *p* ** **-value**	**Non LT *n* = 240,669**	**Yes LT *n* = 5392**	** *p* ** **-value**
Triglycerides/HDL high	17.0 (17.0-17.0)	100.0 (100.0-100.0)	<0.0001	23.4 (23.4-23.4)	100.0 (100.0-100.0)	<0.0001
TyG index high	11.6 (11.6-11.6)	97.5 (96.0-99.0)	<0.0001	25.8 (25.8-25.8)	96.8 (96.0-97.6)	<0.0001
METS-IR high	6.1 (6.1-6.2)	35.1 (33.6-36.6)	<0.0001	10.3 (10.3-10.3)	62.3 (61.5-63.1)	<0.0001

TG/HDL triglycerides/high-density lipoprotein. TyG triglyceride glucose index. METS-IR Metabolic score for insulin resistance. AD atherogenic dyslipidaemia. LT lipid triad. A value of *p* < 0.05 was accepted as a significant difference.

**Table 4 nutrients-15-02105-t004:** Areas under the curve and cut-off points in the different insulin resistance risk scales for predicting atherogenic dyslipidaemia and lipid triad.

	**AD Men**	**LT Men**
	**AUC-Cutoff-Sensib-Specif-Youden Index**	**AUC-Cutoff-Sensib-Specif-Youden Index**
TG/HDL	0.964 (0.964-0.965)-4-0.955-0.916-0.871	0.947 (0.946-0.948)-4.2-0.898-0.882-0.780
TyG index	0.916 (0.914-0.917)-8.9-0.894-0.827-0.721	0.907 (0.905-0.910)-8.9-0.908-0.799-0.707
METS-IR	0.905 (0.903-0.907)-44.5-0.839-0.825-0.664	0.886 (0.883-0.890)-45-0.826-0.800-0.626
	**AD Women**	**LT Women**
TG/HDL	0.991 (0.991-0.991)-3-1.00-0.968-0.968	0.979 (0.978-0.980)-3.1-0.993-0.943-0.936
TyG index	0.974 (0.974-0.975)-8.7-0.969-0.910-0.879	0.963 (0.962-0.965)-8.8-0.927-0.913-0.840
METS-IR	0.872 (0.868-0.876)-40-0.801-0.794-0.595	0.856 (0.849-0.863)-40-0.793-0.776-0.569

TG/HDL triglycerides/high-density lipoprotein. TyG triglyceride glucose index. METS-IR Metabolic score for insulin resistance. AD atherogenic dyslipidaemia. LT lipid triad. AUC area under the curve. Sensib sensibility. Specif specificity.

**Table 5 nutrients-15-02105-t005:** Relations between blood glucose values and lipid profile.

		Women					Men			
Glycaemia		<100 mg/dL	100–125 mg/dL	≥126 mg/dL			<100 mg/dL	100–125 mg/dL	≥126 mg/dL	
	*n*	%	%	%	*p*-Value	*n*	%	%	%	*p*-Value
Cholesterol <200 mg/dL	108,633	91.3	85.4	79.8	<0.0001	147,284	81.3	73.9	67.9	<0.0001
Cholesterol 200–239 mg/dL	47,901	7.7	13.0	17.7		71,274	15.6	22.2	26.8	
Cholesterol ≥240 mg/dL	15,748	1.0	1.6	2.5		27,503	3.1	3.9	5.3	
LDL-c <130 mg/dL	116,109	90.9	85.5	80.6	<0.0001	155,721	80.8	74.7	68.0	<0.0001
LDL-c 130–159 mg/dL	37,234	8.0	13.0	17.2		56,236	16.0	21.8	26.7	
LDL-c ≥160 mg/dL	18,939	1.1	1.5	2.2		34,104	3.2	3.5	5.3	
Triglycerides <150 mg/dL	158,532	89.8	69.7	68.6	<0.0001	187,298	81.2	77.2	63.1	<0.0001
Triglycerides 150–199 mg/dL	9148	9.2	24.5	22.6		30,517	16.5	18.4	27.6	
Triglycerides ≥200 mg/dL	4602	0.9	5.8	8.8		28,246	2.3	4.4	9.3	

LDL-c low-density lipoprotein-cholesterol. A value of *p* < 0.05 was accepted as a significant difference.

**Table 6 nutrients-15-02105-t006:** Multivariate analysis using binary logistic regression between high waist circumference and risk of insulin resistance.

	OR (95% CI)	*p*-Value
<50 years	1	<0.001
≥50 years	1.64 (1.59-1.69)	
Women	1	<0.001
Men	3.03 (2.95-3.12)	
TyG normal	1	<0.001
TyG high	1.25 (1.21-1.28)	
TG/HDL normal	1	<0.001
TG/HDL high	1.87 (1.80-1.95)	
METS-IR normal	1	<0.001
METS-IR high	35.27 (34.15-36.42)	

TG/HDL triglycerides/high-density lipoprotein. TyG triglyceride glucose index. METS-IR Metabolic score for insulin resistance.

## Data Availability

Data are not available due to ethical or privacy restrictions.

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
