# Peer review of "Relationship between Atherogenic Dyslipidaemia and Lipid Triad and Scales That Assess Insulin Resistance"

_nutrients, 2023, doi:10.3390/nu15092105_

Round 1
Reviewer 1 Report
Opinion about the manuscript entitled „RELATIONSHIP BETWEEN ATHEROGENIC DYSLIPIDAEMIA AND LIPID TRIAD AND SCALES THAT ASSESS INSULIN RESISTANCE” sent to Nutrients.
After the reading I am convinced that the text has got the merit and after some improvements it will be widely read by audience. It is interesting and contains line of valuable and useful information. I am working in the field and that text will help me in the future scientific activity.
Remarks:
Line 13 and line 38: “and normal (AD) or high (LT) LDL cholesterol,..” – AD and LT are provided by mistake? You refer AD to Atherogenic dyslipidaemia and LT to lipid triad.
I do not follow the intention to signs of references in the text, not numbers but i, ii, iii, etc. The reference section must be redrafted.
Line 101: you stated that “Triglycerides/HDL-c. Ratio above 2.4 is considered high risk” – the literature provided different approaches, some consider >3 as high risk and some >4. Please add more justification for your >2.4. And <2.4 is no risk? Or moderate ? in what brackets?
Table 1: what p-value refers for? Please add explanation under the table. In the next tables as well, please provide what statistical approach was used and how make interpretation.
Discussion: please expand this section by the opinion as for some others indexes used in the papers related to cardiovascular diseases and diabetes, e.g. HOMA-IR, HOMA-BCF, (TC-HDL)/HDL, HDL profile, etc.
Author Response
Opinión sobre el manuscrito titulado „RELACIÓN ENTRE LA DISLIPIDAEMIA ATEROGÉNICA Y LA TRÍADA DE LÍPIDOS Y LAS ESCALAS QUE EVALUAN LA RESISTENCIA A LA INSULINA” enviada a Nutrients.
Después de la lectura estoy convencido de que el texto tiene el mérito y después de algunas mejoras será ampliamente leído por la audiencia. Es interesante y contiene línea de información valiosa y útil. Estoy trabajando en el campo y ese texto me ayudará en la futura actividad científica.
Observaciones:
Línea 13 y línea 38: “y colesterol LDL normal (AD) o alto (LT),...” – ¿AD y LT se proporcionan por error? Se refiere AD a dislipidemia aterogénica y LT a tríada lipídica.
Fue un error de imprenta, muchas gracias por tu aviso. Hemos procedido a eliminarlos
No sigo la intención de signos de referencias en el texto, no números sino i, ii, iii, etc. La sección de referencia debe ser redactada nuevamente.
Corrected. We appreciate your indication. Possibly the references have been misconfigured in the design. We redraft them. Thank you so much.
Line 101: you stated that “Triglycerides/HDL-c. Ratio above 2.4 is considered high risk” – the literature provided different approaches, some consider >3 as high risk and some >4. Please add more justification for your >2.4. And <2.4 is no risk? Or moderate ? in what brackets?
Following your suggestions, we have added bibliography on the cut-off point and it has been justified in the discussion.
“There is a positive relationship between the TG/HDL-c ratio and the incidence of type 2 diabetes, regardless of initial insulin resistance , . In addition, the TG/HDL-c ratio has been shown to be useful in predicting total mortality, fatal and non-fatal cardiovascular events, regardless of age, race, smoking, hypertension, diabetes, and severity of coronary artery disease , , .
Multiple studies have been published trying to establish the cut-off point for the TG/HDL-c index. Some authors establish the cut-off point of high risk at >3 , other works mark the high risk from >4 , and in other publications the index is established >2.4 , , or even lower , without having been able to establish up to now what the index should be . Being aware that the atherogenic index is related to the number of small LDL parti-cles, and that a TG/HDL-c index >2 indicates a greater number of small LDL particles . In our work we have used >2.4 as the high risk cut-off point, which is the cut-off point of other studies carried out in the Hispanic populationlvii,lviii, and the cut-off point for the Spanish population recommended by our scientific societies, semFYC (Spanish Society of Medicine). Family and Community), SEMERGEN (Spanish Society of Primary Care Physicians), SEMG (Spanish Society of General Medicine) and SEA (Spanish Society of Arteriosclerosis) . Below 2.4, we could consider moderate risk.”
Table 1: what p-value refers for? Please add explanation under the table. In the next tables as well, please provide what statistical approach was used and how make interpretation.
Explanation of p value added below tables. Very thankful
Discussion: please expand this section by the opinion as for some others indexes used in the papers related to cardiovascular diseases and diabetes, e.g. HOMA-IR, HOMA-BCF, (TC-HDL)/HDL, HDL profile, etc.
Following their recommendations, we have expanded with a paragraph the justification of the indices used against HOMA-IR, QUICKI, Matsuda and others.
“De las diferentes escalas de resistencia a la insulina aceptadas, hemos seleccionado Triglicéridos/HDL-c, índice TG y METS-IR. Existen otros métodos reconocidos para determinar la resistencia a la insulina, QUICKI, HOMA-IR, índice de Matsuda, sin embargo es necesario conocer el nivel de insulina en ayunas, el cual no teníamos en nuestro estudio y además es muy difícil de obtener en médicos de atención primaria ' oficinas. De las fórmulas no basadas en insulina, hemos seleccionado aquellas que han mostrado una mejor correlación con la resistencia a la insulina, .“
Agradecemos sus consejos y recomendaciones. Hemos procedido a realizar todos los cambios sugeridos. Confiamos en que serán de su agrado.

Reviewer 2 Report
The authors have studied a marvelous database consisting of 418,343 workers (172,282 women and 246,061 men) from different regions in Spain with comprehensive lipid data on triglycerides, LDL, and HDL. They have anthropometric data which was evaluated in a prior Nutrients publication:
Ramírez-Manent JI, Jover AM, Martinez CS, Tomás-Gil P, Martí-Lliteras P, López-González ÁA. Waist Circumference Is an Essential Factor in Predicting Insulin Resistance and Early Detection of Metabolic Syndrome in Adults. Nutrients. 2023; 15(2):257. https://doi.org/10.3390/nu15020257
They have liver enzyme data as well. With this outstanding database, they can analyze numerous factors creating a landmark series. In this particular article, they have chosen to analyze various scales associated with insulin resistance:
- Triglycerides/HDL-c. Ratio above 2.4 is considered high riskxxviii. 101
- Triglyceride-glucose index (TG index)xxix, obtained by applying the following for- 102
mula Ln(Triglycerides [mg/dL] x glucose [mg/dL]/2). Values above 8.8 are considered 103
high. 104
Other parameters were obtained from the TyG index: TyG-BMIxxx, TyG-waist30, and 105
TyG-WtHRxxxi (waist to height ratio). 106
- Metabolic score of insulin resistance (METS-IR)xxxii 107
METS-IR = Ln((2*Glucose)+Triglycerides)*BMI)/(Ln(HDL-c)). High values are con- 108
sidered as 50 and over. 109
The key word here is "associated with". It is clear that there is a major association of lipid abnormality with insulin resistance. There is also a clear association of insulin resistance with overweight. The key word here again. is "association". Of course, the authors realize that they lack actual insulin levels in their database. They are invoking associations with associations and that is not unacceptable. It is just uninteresting. They really have no convincing measure of insulin resistance in this article.
I am not recommending rejection, but rather encouraging the authors to use their magnficent resources to accomplish something of true value. Here is a simple suggestion that might pique the interest of the reader: How about showing the correlation of their lipid measures with fasting glucose? Their Met-IRS is supported by one publication:
Omar Yaxmehen Bello-Chavolla, Paloma Almeda-Valdes, Donaji Gomez-Velasco, Tannia Viveros-Ruiz, Ivette Cruz-Bautista, Alonso Romo-Romo, Daniel Sánchez-Lázaro, Dushan Meza-Oviedo, Arsenio Vargas-Vázquez, Olimpia Arellano Campos, Magdalena del Rocío Sevilla-González, Alexandro J Martagón, Liliana Muñoz Hernández, Roopa Mehta, César Rodolfo Caballeros-Barragán, Carlos A Aguilar-Salinas, METS-IR, a novel score to evaluate insulin sensitivity, is predictive of visceral adiposity and incident type 2 diabetes, European Journal of Endocrinology, Volume 178, Issue 5, May 2018, Pages 533–544
Again, associations with glucose clamp results and with HOMA-IR. The key word is association.
Much more interesting would be the correlation between the lipid scales they are using and impaired fasting glucose in their population. Perhaps they even have subjects with glucose values in the diabetes range.
If they could lift their game plan, their wonderful study could certainly achieve the recognition it deserves.
Author Response
Reviewer 2
The authors have studied a marvelous database consisting of 418,343 workers (172,282 women and 246,061 men) from different regions in Spain with comprehensive lipid data on triglycerides, LDL, and HDL. They have anthropometric data which was evaluated in a prior Nutrients publication:
Ramírez-Manent JI, Jover AM, Martinez CS, Tomás-Gil P, Martí-Lliteras P, López-González ÁA. Waist Circumference Is an Essential Factor in Predicting Insulin Resistance and Early Detection of Metabolic Syndrome in Adults. Nutrients. 2023; 15(2):257. https://doi.org/10.3390/nu15020257
They have liver enzyme data as well. With this outstanding database, they can analyze numerous factors creating a landmark series. In this particular article, they have chosen to analyze various scales associated with insulin resistance:
- Triglycerides/HDL-c. Ratio above 2.4 is considered high riskxxviii. 101
- Triglyceride-glucose index (TG index)xxix, obtained by applying the following for- 102
mula Ln(Triglycerides [mg/dL] x glucose [mg/dL]/2). Values above 8.8 are considered 103
high. 104
Other parameters were obtained from the TyG index: TyG-BMIxxx, TyG-waist30, and 105
TyG-WtHRxxxi (waist to height ratio). 106
- Metabolic score of insulin resistance (METS-IR)xxxii 107
METS-IR = Ln((2*Glucose)+Triglycerides)*BMI)/(Ln(HDL-c)). High values are con- 108
sidered as 50 and over. 109
The key word here is "associated with". It is clear that there is a major association of lipid abnormality with insulin resistance. There is also a clear association of insulin resistance with overweight. The key word here again. is "association". Of course, the authors realize that they lack actual insulin levels in their database. They are invoking associations with associations and that is not unacceptable. It is just uninteresting. They really have no convincing measure of insulin resistance in this article.
We have proceeded to change the word relationship to association, as you suggest. Thank you so much.
I am not recommending rejection, but rather encouraging the authors to use their magnficent resources to accomplish something of true value. Here is a simple suggestion that might pique the interest of the reader: How about showing the correlation of their lipid measures with fasting glucose? Their Met-IRS is supported by one publication:
Omar Yaxmehen Bello-Chavolla, Paloma Almeda-Valdes, Donaji Gomez-Velasco, Tannia Viveros-Ruiz, Ivette Cruz-Bautista, Alonso Romo-Romo, Daniel Sánchez-Lázaro, Dushan Meza-Oviedo, Arsenio Vargas-Vázquez, Olimpia Arellano Campos, Magdalena del Rocío Sevilla-González, Alexandro J Martagón, Liliana Muñoz Hernández, Roopa Mehta, César Rodolfo Caballeros-Barragán, Carlos A Aguilar-Salinas, METS-IR, a novel score to evaluate insulin sensitivity, is predictive of visceral adiposity and incident type 2 diabetes, European Journal of Endocrinology, Volume 178, Issue 5, May 2018, Pages 533–544
Again, associations with glucose clamp results and with HOMA-IR. The key word is association.
Much more interesting would be the correlation between the lipid scales they are using and impaired fasting glucose in their population. Perhaps they even have subjects with glucose values in the diabetes range.
If they could lift their game plan, their wonderful study could certainly achieve the recognition it deserves.
We appreciate all your recommendations and following your instructions. We have created a new table (TABLE 5) that stratified fasting blood glucose levels (<100 mg/dL,100-125 mg/dL,, ≥126 mg/dL) by gender with the different lipid values used. also stratified. And commented in the discussion.
“In the same way that we have evaluated the inclusion of lipids in insulin resistance formulas, we have considered it interesting to evaluate the relationship between the different lipid fractions of plasma and fasting glycemia. Since the transformation of excess glucose into lipids has been demonstrated, and the relationship of elevated glucose levels with diabetes mellitus and insulin resistance .
In our work we found that as total cholesterol figures increase from below 200 mg/dL to more than 240 mg/dL, LDL-c from below 130 mg/dL to more than 160 mg/dL, or Triglycerides from less than 150 mg/dL to more than 200 mg/dL increase fasting blood glucose levels in both men and women. Which agrees with other published studies , . This coincides with the results obtained, since both triglycerides and fasting glucose intervene in the TyG index and the METS-IR formula. The latter is already recognized as a useful tool to assess metabolic health in primary carexxxii.
The influence of lipids on the development of arteriosclerotic disease is known . Which produces an inflammation of the arterial wall with accumulation of lipids in it .
The presence of elevated plasma levels of LDL-c constitute a cardiovascular risk factor . Diabetes mellitus has also been shown to accelerate the development of cardiovascular disease through hyperglycemia, which, together with elevated LDL-c levels, leads to increased oxidative stress and inflammation .
In the patient with diabetes, the change in the serum lipid profile causes a greater production of triglycerides in the liver. At the same time, the lack of insulin causes an increase in the production of ROS (reactive oxygen species) and chronic inflammation that consequently accelerates the development of arteriosclerosis and cardiovascular disease . These effects are favored by insulin resistance .
In our work, we can see how there is an association between the increase in blood lipids and fasting serum glucose levels. Similarly, we found an association between atherogenic dyslipidaemia and lipid triad with the resistance risk scales evaluated.”
We appreciate your advice and recommendations. We have proceeded to make all suggested changes. We trust that they will be to your liking.

Round 2
Reviewer 2 Report
The authors are now moving to address the major problem with this paper: It uses several noninsulin based measurements of lipids as if that were conclusive of insulin resistance. There are a handful of papers which attempt to portray these lipid measurements as significant. By no means are these scales accepted as yet. In this paper, the authors show that there are interrelationships of their lipid measurements associating the noninsulin based measures of insulin resistance with correlative non imaging based measures of atherogenesis. That is somewhat interesting, but is a poor use of their wonderful database. They have now added measurements of fasting glucose to their analysis. That now begins to be interesting. They do not have insulin levels, but they do have the ability to measure impaired fasting glucose and, indeed, the possible presence of diabetes, using the surrogate measure of fasting glucose greater than 125 mg/dl. Of course fasting glucose is as much a measure of deficient insulin secretory response as of insulin resistance, and they have no measure of insulin secretion. Nonetheless if they can show the degree of correlation of their three chosen lipid scales fasting glucose, that is a valuable contribution. To go further, they must show the numbers of patients in each of the cohorts in their new table. Finally, to make this paper truly exciting, they can do a logistic regression with their non-insulin based scales and waist circumference (which they previously published) as predictive of diabetes as indicated by fasting glucose greater than 125 mg/dl to show the comparative regression coefficients.. Now, if they can do that, it will go far toward establishing the value of their analyses. These authors have an extraordinary database for future studies. They should do all that they can to make each publication meaningful and generate entthusiasm.
Author Response
Dear reviewer,
We have tried to respond to your recommendations. For which we have performed a logistic binary multivariate analysis between the waist circumference and the insulin resistance formulas evaluated.
We trust that we have responded to what you requested. Thank you so much.